

# A REtrieval Method for optical and physical Aerosol Properties in the stratosphere (REMAPv1)

Andrin Jörimann[1,2], Timofei Sukhodolov[1], Beiping Luo[3], Gabriel Chiodo[3,4], Graham Mann[5,6], and Thomas Peter[3]

[1]Physikalisch-Meteorologisches Observatorium Davos / World Radiation Center, Davos, Switzerland
[2]Institute for Particle Physics and Astrophysics, ETH Zürich, Zürich, Switzerland
[3]Institute for Atmospheric and Climate Science, ETH Zürich, Zürich, Switzerland
[4]Institute of Geosciences, Spanish National Research Council, Madrid, Spain
[5]School of Earth and Environment, University of Leeds, Leeds, UK
[6]UK National Centre for Atmospheric Science, University of Leeds, Leeds, UK

**Correspondence:** Andrin Jörimann (andrin.joerimann@pmodwrc.ch)

**Abstract.** The stratospheric aerosol is an important climate forcing agent as it scatters some of the incoming solar radiation back to space, thus cooling the Earth's surface and the troposphere. At the same time it absorbs some of the upwelling terrestrial radiation, which heats the stratosphere. It also plays an important role in stratospheric ozone chemistry by hosting heterogeneous reactions. Major volcanic eruptions can cause strong perturbations of stratospheric aerosol, changing its radiative and chemical effects by more than an order of magnitude. Many global climate models require prescribed stratospheric aerosol

as input to properly simulate both climate effects in the presence and absence of volcanic eruptions. This paper describes REMAP, a retrieval method and code for aerosol properties that has been used in several model intercomparison projects (under the name SAGE-3/4$\lambda$). The code fits a single-mode log-normal size distribution for a pure aqueous sulfuric acid aerosol to aerosol extinction coefficients from observational or model data sets. From the retrieved size distribution parameters the code

calculates the effective radius, surface area density, as well as extinction coefficients, single-scattering albedos and asymmetry factors of the aerosol within the wavelength bands specified by individual climate models. We validate the REMAP using balloon observations after the Pinatubo and Hunga-Tonga Hunga Ha'apai volcanic eruptions, as well as four decades of LIDAR measurements. Within the constraints of a single-mode log-normal distribution REMAP generates realistic effective radii and surface area densities after volcanic eruptions and generally matches the LIDAR backscatter time series within measurement

uncertainty. Deviations in aerosol backscatter up to a factor of 2 arise when (non-volcanic) tropospheric intrusions (e.g. from wildfires) are present and the size distribution deviates significantly from the single-mode log-normal type. We describe the products that have been used in CCMI, CMIP6 and other model intercomparison projects, and provide practical instructions for use of the code.





## 1 Introduction

The stratospheric aerosol has been of scientific interest since its discovery by Christian Junge and colleagues in 1960 (Junge et al., 1961; Chagnon and Junge, 1961). Aerosol particles have an impact on the radiative balance of the Earth system (Robock, 2000; Solomon et al., 2011; Santer et al., 2014) and they host important heterogeneous chemical reactions (Hofmann and Solomon, 1989; Solomon, 1999; Tilmes et al., 2008). Therefore, global models require information on stratospheric aerosol

records and climatologies.//

The strength of heterogeneous chemical reactions scales with the surface area density (SAD) of stratospheric aerosols. The $N_2O_5$ hydrolysis on aerosol surfaces (e.g. Robinson et al. (1997)) alters the partitioning of active nitrogen $NO_x$ to total reactive nitrogen $NO_y$, which affects stratospheric ozone chemistry throughout the aerosol layer, especially under volcanically perturbed conditions (Fahey et al., 1993). In very cold conditions, such as in the winter polar vortex or in the vicinity of the

tropical tropopause, additional heterogeneous reactions may activate chlorine from its reservoir species, leading to enhanced ozone depletion (Hanson et al., 1994; Solomon, 1999; Tilmes et al., 2008). These chlorine activating reactions occur mainly at the liquid-gas interface and to a lesser extent within the liquid bulk phase of these cold binary $H_2SO_4$-$H_2O$ droplets. Therefore, size information is also required for the corresponding modelling of the heterogeneous reactions, in addition to the SAD (Hanson et al., 1994). Besides these cold $H_2SO_4$-$H_2O$ droplets, polar winter conditions also favor the formation of polar strato-

spheric clouds (PSCs), i.e. droplets and solid hydrate particles containing $HNO_3$. However, PSCs are not the subject of this article and the transition from cold binary $H_2SO_4$-$H_2O$ particles to large ternary $HNO_3$-$H_2SO_4$-$H_2O$ droplets and hydrates must be treated individually be each global climate model.

The radiative forcing of the stratospheric aerosol has been addressed by many authors in particular with respect to volcanic

perturbations (Russell et al., 1993; Sato et al., 1993; Stenchikov et al., 1998; Robock, 2000; Stothers, 2001; Ammann et al., 2003; Thomason and Peter, 2006; Thomason et al., 2008; Vernier et al., 2011; Solomon et al., 2011; Arfeuille et al., 2013, 2014; Santer et al., 2014; Zanchettin et al., 2016). Recent major volcanic eruptions – e.g. El Chichón in 1982 and Pinatubo in 1991 – led to a temporary decrease of global mean surface temperature, estimated to reach up to 0.5 K (Santer et al., 2014). The moderate volcanic eruptions since 2005 of Augustine, Soufrière Hills, Shiveluch, Kasatochi, Sarychev and Nabro led to a mod-

erate, but persistent increase in the stratospheric aerosol (Vernier et al., 2011; Carn et al., 2017), which may partly explain the early 21st century 'warming hiatus' (Solomon et al., 2011; Santer et al., 2014; Andersson et al., 2015). Stratospheric aerosol continues to be elevated by volcanic eruptions (Ambrae, Ulawun, Raikoke) in the past few years (Carn, 2022), including the water-rich Hunga Tonga-Hunga Ha'apai (HTHH) eruption (Vömel et al., 2022), joined by non-sulfuric sources: several large wildfire events in British Columbia, Australia and Colorado (Madhavan et al., 2022; Trickl et al., 2023). A precise description

of the radiative properties of the stratospheric aerosol following such events requires understanding of the distribution and transformation of particles, i.e. size distribution information, in the fresh volcanic plume (e.g. Sheng et al. (2015)). Global size distribution data sets, however, do not exist from direct measurements and need to be retrieved from satellite measured radiative properties. Thus, their quality relies on the retrieval method. This type of data could further inform speculation, as



to whether artificial stratospheric SO₂ injections are a suitable candidate for partially counteracting global warming caused by
greenhouse gas emissions (e.g. Crutzen (2006); Heckendorn et al. (2009); Weisenstein et al. (2022)).

The operation of global climate models requires knowledge on the radiative forcing by the stratospheric aerosol. This aerosol forcing can either be calculated online, provided that the employed model has microphysics and chemistry modules or it can be prescribed as external forcing, provided a sufficiently good observational data set is available. The former requires to model
the conversion of aerosol precursor gases (OCS, SO₂) via photolysis and oxidation reactions into gaseous $H_2SO_4$, which can condense on preexisting aerosol particles or nucleate new ones; calculate size distributions of particles and transport them throughout the stratosphere; and finally calculate by Mie theory the backward scattering of visible solar radiation and the reduction of the incoming solar energy. The extensive physico-chemical modelling is computationally intensive, but is now used by a number of the global climate models (Timmreck et al., 2018; Brodowsky et al., 2024). Yet, the resulting aerosol forcing
may differ significantly over different models with interactive aerosols (Clyne et al., 2021; Quaglia et al., 2023), representing a major source of uncertainty. However, most Global Circulation Models (GCMs) still rely on prescribed chemical and aerosol forcing fields, and even many Chemistry-Climate Models (CCMs) prescribe stratospheric aerosols as a compromise between models' complexity and computational efficiency. Thus, a method of establishing the radiative forcing fields of the stratospheric aerosol for the use in global climate models is required. The operation of these models typically requires the
space-time-resolved data fields of aerosol extinction coefficients (AEC in text; $\beta$ in mathematical notation), single-scattering albedos (SSA; $\omega$), and asymmetry factors (AF; $g$) for each wavelength band in each of the models. Establishing these fields, together with SAD and size information, is a major prerequisite for model intercomparison projects (MIPs) (Lanzante and Free, 2008; Eyring et al., 2010; Gettelman et al., 2010; Eyring et al., 2016; Morgenstern et al., 2017).

The starting point for calculating the radiative properties of aerosols is a size distribution that characterizes an aerosol ensemble. A method to derive size distribution parameters from SAGE (Stratospheric Aerosol and Gas Experiment) II multi-wavelength aerosol extinction measurements was originally proposed by Yue (1986) and Yue et al. (1986). They used AEC ratios to solve a system of equations for the parameters of a single-mode log-normal (SLN) size distribution. More efforts to retrieve SLN size distributions followed (e.g. Stenchikov et al. (1998) and references therein, Bingen et al. (2003); Arfeuille
et al. (2013); Wrana et al. (2021)). In quiescent periods the SLN size distribution approximates the stratospheric aerosol fairly well (Deshler et al., 1992; Wurl et al., 2010). However in volcanically enhanced stratospheric aerosol additional modes are often present (Deshler et al., 1992; Pusechel et al., 1992). Russell et al. (1996) mention that retrieved SLN size distributions can still produce accurate aerosol properties – even when the aerosol consists of two modes – given that the real distribution is not "extremely bimodal" (i.e. the modes overlap sufficiently well and do not have similar number densities). However,
von Savigny and Hoffmann (2020) warn of systematic biases caused by the SLN size distributions retrieved from occultation measurements. These biases apply to conditions of the stratospheric aerosol, when a SLN size distribution is insufficient in capturing the real distribution. While Thomason et al. (2008) retrieved aerosol SAD for two monodisperse modes with prescribes number densities, most observational products do not provide enough independent information (wavelength channels) to rea-





sonably constrain all parameters of a dual-mode log-normal (DLN) size distribution. In this paper we focus on a generalized
best-fit retrieval approach for SLN size distributions and discuss the limitations of the SLN approximation using comparisons
with balloon-borne in situ and LIDAR measurements.

Dedicated satellite missions to observe the stratospheric aerosol started in 1979 with the SAGE program. Spanning two full
decades (1984-2005), the SAGE II data is a cornerstone of stratospheric aerosol observations, which has been used exten-
sively. It includes AEC at the wavelengths 1020, 525, 452 and 386 nm, providing information on the size of aerosol particles.
However, caution is advised, as the uncertainties in these 4 channels differ considerably, with systematically higher standard
deviations in channels 452 and 386 nm. The latest release of SAGE II data is version 7 (Damadeo et al., 2013) with improved
retrieval algorithms compared to earlier versions. In June 2017 the operational period of the Stratospheric Aerosol and Gas Ex-
periment III mounted externally on the International Space Station (SAGE III on ISS) started providing the SAGE III aerosol
extinction on 9 different wavelengths, continuing the record of the SAGE instrument family after an interruption of over a
decade. Besides the improved range of the wavelength range further into the near-infrared (the longest wavelength of SAGE
III is 1544 nm, as compared to its predecessor SAGE II with 1020 μm), a cloud-cleared version of the SAGE III data has been
introduced. The use of an aerosol/cloud categorization algorithm (Kovilakam et al., 2023) eliminates a time-intensive, pre-
viously manual step in pre-processing. The Global Space-based Stratospheric Aerosol Climatology (GloSSAC) incorporates
both the SAGE II and SAGE III data sets, as well as CLAES-ISAMS (Cryogenic Limb Array Etalon Spectrometer – Improved
Stratospheric and Mesospheric Sounder, Lambert et al. 1997), HALOE (Halogen Occultation Experiment, Hervig et al. 1995),
OSIRIS and CALIOP data (described in S1.3) to produce a continuous AEC product from 1979 onward (see supplementary
materials for more information on data sets). Since 2012, the Ozone Mapping and Profiler Suite Limb Profiler (OMPS-LP)
onboard the Suomi NPP satellite provides limb backscatter measurements. From these data, Taha et al. (2021) have retrieved
AEC on 6 wavelengths, however, they report limited accuracy in some of the channels. In the future, OMPS-LP data might be
incorporated into GloSSAC as well, potentially making the composite more robust. Additionally, a new Climate Data Record
of Stratospheric Aerosols (CREST) was published recently (Sofieva et al., 2024), providing a single-wavelength composite of
6 limb scatter and occultation measurement data sets.

115 In this paper we describe and validate the latest version of a generalized method to retrieve SLN size distributions for
sulfuric aerosol from the above-mentioned AEC data sets and use these to calculate a variety of aerosol optical and mass-
related properties. The method is termed REtrieval Method for Aerosol Properties (REMAP). Its strength lies in its flexibility
to provide optical properties for any prescribed wavelength band, so the output can be tailored to the spectral resolution of
any radiation scheme. The REMAP method, previously termed SAGE-4$\lambda$ and SAGE-3$\lambda$, has been used in multiple past and
120 ongoing MIPs (Table 1), but has not yet been described in sufficient detail. Arfeuille et al. (2013) described the preparation
of a Pinatubo forcing data set using this method, however this was only a brief description of one specific use case. Here,
we start by giving an overview of all the projects REMAP was used in and the resulting data sets. We then describe the





general methodology in full and give practical information on how to use the code, followed by a product validation against measurements.

## 2 Product overview

REMAP was first used to create the aerosol forcing for global models within the IGAC/APARC Chemistry-Climate Model Initiative (CCMI) Phase 1 experiment (Eyring et al., 2013). To this end, different satellite data sets were selected, screened for cloud contamination, interpolated and ultimately combined to create a continuous and consistent 50 year record (1960 – 2011). This record was termed SAGE-4λ, because its backbone were the four SAGE II wavelengths (4λ). In addition to SAGE II data, some other satellite AEC, sun photometer data, as well as information from tropical ground-based LIDARs for the filling of data gaps under volcanically opaque conditions were combined. For details about the SAGE-4λ record, including gap filling, see S1. In preparation for CMIP6, Thomason et al. (2018) constructed the first version of GloSSAC (v1.0), which contained an error in the conversion of CLAES data to SAGE II wavelengths. This was corrected in v1.1 as described in Kovilakam et al. (2020), which we used for the CMIP6 forcings. We based this data set on the homogenized single product GloSSAC, rather than incorporating a variety of different sources, like we did for CCMI-1. Still, the CMIP6 forcing data set is primarily based on SAGE II. However, we did not use the shortest SAGE II wavelength at 386 nm, since its quality was insufficient to improve the result. The data set was thus termed SAGE-3λ. Revell et al. (2017) describe the CLAES/SAGE II conversion error in their Corrigendum and show differences in the aerosol loading and forced temperature and ozone anomalies that arise from using the improved SAGE-3λ data set versus the previous SAGE-4λ. Shortly after we released SAGE-3λ to the modelling community, GloSSAC was further updated with new and improved satellite data sets as GloSSACv2.0, described in Kovilakam et al. (2020). It also extended to the end of 2018 and was subsequently used for the historical hindcast period of the CCMI Phase 2 (CCMI-2022) modelling activity (Plummer et al., 2021), which required prescribed radiative forcing data sets for stratospheric aerosol. Another set of sensitivity runs within CCMI-2022 simulated a climate intervention scenario and was called senD2-sai. For this experiment, the forcing needed to be derived from a separate model simulation – instead of observations – and spectrally regridded for participating models. Details on this procedure are given in Sect. 3.5. Finally, the latest version of GloSSAC at the time of writing (v2.22) includes an aerosol/cloud categorization algorithm (Kovilakam et al., 2023) and extends to the end of 2023. We used all available GloSSAC wavelengths with their associated standard deviations (eliminating the quality reduction of the shortest wavelength by itself) for the Hunga Tonga-Hunga Ha'apai Impact Model Observations Comparison (HTHH-MOC) forcings (Zhu et al., 2024).

Table 1 shows all the data sets produced with REMAP that were distributed to modelling groups, along with information about the respective inputs. To summarize, REMAP was used to provide data of stratospheric aerosol optical properties to

- 9 models participating in CCMI-1 in support of ozone and climate assessments (Eyring et al., 2013; Hegglin et al., 2014) [SAGE-4λv2 data set (Luo, 2013)],



**Table 1.** Data sets produced with REMAP and used in different model intercomparison projects to prescribe a uniform stratospheric aerosol.

| Data set | time | MIP | # of models | input data set | boundary conditions (T & RH) |
|---|---|---|---|---|---|
| SAGE-4λv2 | 1960-2011 | CCMI-1 | 9 | various∗ | ERA-Interim – 1991 climatology |
| SAGE-3λv4 | 1850-2014 | CMIP6 | >20 | GloSSACv1.1 | ERA-Interim – 1991 climatology |
| REMAP-CCMI-2022-ref | 1960-2018 | CCMI-2022 | 12 | GloSSACv2.0 | ERA-Interim – 1991 climatology |
| REMAP-CCMI-2022-sai | 2025-2100 | CCMI-2022 | 4 | WACCM output | transient WACCM output |
| REMAP-GloSSAC-2023 | 1979∗-2023 | HTHH-MOC | 7 | GloSSACv2.22 | transient ERA5 |

∗Even though the data set was established for the entire duration of the GloSSACv2.22 data set (1979-2023), this MIP only required forcing from 2019-2023.

- some 20 models participating in the Coupled Model Intercomparison Project Phase 6 (CMIP6) to better understand past, present and future climate in a multi-model context (Eyring et al., 2016) [SAGE-3λv4 data set (Luo, 2017)],

   - 12 models participating in the CCMI Phase 2 / CCMI-2022 historical runs (also following CMIP6 forcing guidelines) (Plummer et al., 2021) [REMAP-CCMI-2022-ref data set (Luo, 2020)],

   - 4 models participating in the CCMI-2022 senD2-sai experiment to assess a climate intervention (CI) scenario (Plummer
et al., 2021) [REMAP-CCMI-2022-sai data set (Jörimann, 2023); more information in Sect. 3.5],

   - 7 models participating in the Hunga Tonga-Hunga Ha'apai Impact Model Observations Comparison (HTHH-MOC) to examine the effects of the largest phreatomagmatic explosion in the satellite record (Zhu et al., 2024) [REMAP-GloSSAC-2023 dataset (Jörimann, 2024)].

## 3   Description of the REtrieval Method for Aerosol Properties (REMAP)

In order to estimate the radiative properties of the aerosol particles, the size distribution is required. It can be retrieved, in a best-fit sense, by comparing the calculated AEC of all size distributions in the allowed parameter space against observations. With the size distribution characterized, the AEC, SSA and AF of the aerosol ensemble can be calculated using Mie theory (Bohren and Huffman, 1998). To this end, we assume that the stratospheric aerosol comprises pure aqueous sulfuric acid solution droplets. The sulfuric acid concentration is calculated from the atmospheric relative humidity and temperature using
the water vapor pressure of aqueous sulfuric acid (Luo et al., 1995; Carslaw et al., 1995). For the relative humidity (RH) and temperature, both model output or reanalysis data can be used. The refractive index as a function of temperature and $H_2SO_4$ concentration is then calculated using the parameterization of Luo et al. (1996) for visible wavelengths and measurements of Biermann et al. (2000) for infrared wavelengths. In this way, we take the concentration and temperature dependence of the refractive index into account, both in the size distribution retrieval procedure and also in the final calculation of the optical
properties.



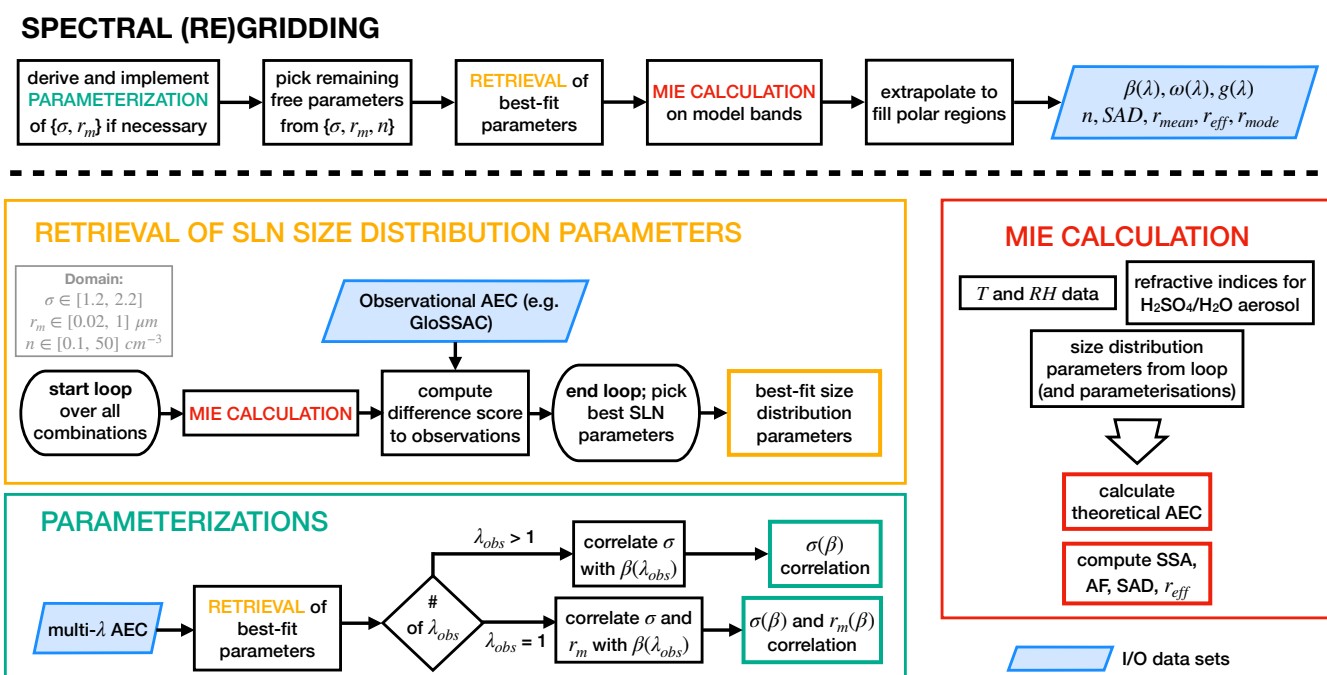

**Figure 1.** Scheme of the REMAP algorithms in general form. The spectral regridding method always involves the retrieval (yellow box) of aerosol size distribution parameters $(n, r_m, \sigma)$ from observational AEC data and the Mie calculation (red box) of radiative properties (aerosol extinction coefficient $\text{AEC}(\lambda)$, single-scattering albedo $\text{SSA}(\lambda)$, asymmetry factor $\text{AF}(\lambda)$) in spectral resolution of choice. Optionally, the retrieval procedure is improved and in some cases made possible in the first place by parameterizations (turquoise box) separately retrieved from a data set with a high enough number of channels (multi-$\lambda$ data set). For details see text.

## 3.1 Retrieval of the size distribution

Figure 1 summarizes REMAP in its most general form and latest state. The core element is the retrieval procedure (yellow box), which uses the Mie calculation (red box) and is itself used in the optional derivation of parameterizations (turquoise box). The retrieval procedure operates on a data set of AEC either from measurements or the output of a model simulation, which here we call the observational AEC data set.

The basic assumption throughout the whole method is that the aerosol size distribution of the aerosols can be represented by a SLN particle size distribution:

$$\frac{dn(r)}{dr} = n\frac{1}{\sqrt{2\pi}\, r \ln^2 \sigma} \exp\left(-\frac{\ln^2(r/r_m)}{2\ln^2 \sigma}\right) \tag{1}$$





There are three unknowns: the number density, $n$, the mode radius, $r_m$, and the half-width $\sigma$ of the aerosol SLN size distribution. Ideally, the retrieval procedure fits all three parameters to the observational AEC. In that case the parameters can be directly retrieved by picking the parameter combination that best reproduces the measured AEC. However, this requires data on at least 3 wavelengths of good quality (see Sect. 3.2 for what constitutes good quality). If there is no complete coverage with at least 3 wavelengths of good quality, or to simply improve the ouput consistency (cf. Arfeuille et al. (2013), parameterizations

can be used to replace one or two parameters. Here, we describe the method systematically for the distinct cases of $\geq 3$, 2 and 1 available wavelength.

(a) **3 or more wavelengths**

The three parameters $n$, $r_m$ and $\sigma$, which fit best to the measured AEC are directly retrieved (*full retrieval*, yellow box in Figure 1). The best fit parameters are selected based on the minimum difference score $D$ between measured and

theoretical AEC across the range of possible parameter values. To achieve this, an program goes through the following steps for each set of parameters in the domain (grey box):

- The theoretical AEC $k_{theor}$ are calculated on the same wavelengths $\lambda$ that are available in the observational data using Mie theory (red box).

- For each wavelength the difference score $D$ of the calculated to the measured extinction coefficients $k_{obs}$ is com-

puted.

- If available, the standard deviation $\varsigma$ of the measurements is also taken into account. The theoretical-to-observed AEC ratio for each wavelength is weighted inversely with the associated scatter (i.e, the standard deviation), which is typically higher at smaller wavelengths due to higher molecular-to-aerosol extinction ratio. If no standard deviations are available, each wavelength is weighted the same.

- The ratios for each wavelength are summed up to a total (weighted) difference score. The program keeps track of the lowest achieved difference score and the corresponding parameters $n$, $r_m$ and $\sigma$. It only updates the parameters, once a lower difference score is generated. After all combinations are run, the best fit (according to Mie theory and assuming a SLN size distribution) is found.

- Summarized in mathematical notation, the difference score

$$D = \sum^{\lambda} \left[ \ln\left( \frac{\beta_{theor}(\lambda)}{\beta_{obs}(\lambda)} \right) \cdot weight \right]^2, \quad weight = \frac{\frac{\beta(\lambda)}{\varsigma(\lambda)}}{\sqrt{\sum^{\lambda} \left( \frac{\beta(\lambda)}{\varsigma(\lambda)} \right)^2}} \quad (2)$$

is minimized at each grid point. From this difference score, we also report an error $E$, given by

$$E = \sqrt{D}. \quad (3)$$

The error $E$ typically assumes its lowest values in the core of the Junge layer. This is illustrated in Figure 2, which is a latitude-altitude map of the mean error over 5 years that came from a retrieval on 6 wavelengths of



**Figure 2.** Latitude-altitude plot of the 5-year mean error from a retrieval of 6 SAGE III wavelengths. The black line indicates the average tropopause height. The Junge layer is discernible from the minimum error values. The distinct rise in error above 27 km at all latitudes comes from the observational data set GloSSACv2.22 itself, which includes a high-altitude climatology.

SAGE III data. The error tends to be higher above 27 km, due to those data coming from a climatology and in the lowest stratospheric region, just above the tropopause. The actual values depend on the number of wavelengths, the availability of measurement standard deviation and the strength of the AEC signal itself. Therefore, it cannot be used to compare the retrieval quality between retrievals, where any of these factors vary, only within a single run.

To improve the convergence of the algorithm we constrained the range of $\sigma$ to $1.2 - 2.2$, the range of $r_m$ to $0.02 - 1$ μm
and the range of $n$ to $0.1 - 50$ cm$^{-3}$. These upper and lower limits are physically reasonable constraints derived from the long-term in situ measurements over Laramie, Wyoming (Deshler, 2008). For the Mie calculation, the environmental variables temperature and RH are required, besides the size distribution. Temperatures and RH are also required to calculate the aerosol refractive index (Luo et al., 1996; Biermann et al., 2000), the third boundary condition for the



calculation of theoretical aerosol optical properties. Table 1 shows, where these boundary conditions came from for
different REMAP data sets.

The results of a *full retrieval* can still be heterogeneous, when all three parameters are unconstrained within the permitted
parameter space. To improve this, a parameterization (see turquoise box in Figure 1) lays a further constraint on the
retrieval process. With the parameterization function of the form $\sigma(k_{param})$ the retrieval algorithm can be repeated, this
time only fitting the remaining two parameters.

(b) **2 wavelengths**

If only two different wavelengths are available for the retrieval, a *full retrieval* cannot be performed and a parame-
terization function is required to start the process. These functions make use of an empirical relationship of one size
distribution parameter with the observational AEC data, thus eliminating one degree of freedom. To establish the empir-
ical relationship, another data set is required that is apt for a *full retrieval*. We found that the distribution half-width $\sigma$
and mode radius $r_m$ are best suited for parameterizations, since they most reliably produce monotonic parameterization
functions. Figure 3 shows a 2D histogram of the SAGE III extinction coefficients at 1022 nm and the retrieved $\sigma$ above
20 km. We used AEC on the wavelength 1022 nm, because they were very well reproduced by the algorithm (cf. Figure
4). An eighth grade fourier fit to the median (red line in Figure 3) was used to parameterize $\sigma$ in this case:

$$
\sigma = \begin{cases}
2.19 & \text{if } \ln\beta_{1022} < -13, \\[4pt]
\begin{aligned}
1.636 \\
&-0.09632\cos(\ln\beta_{1022}\cdot 0.6463) - 0.4427\sin(\ln\beta_{1022}\cdot 0.6463) \\
&-0.08697\cos(2\ln\beta_{1022}\cdot 0.6463) + 0.1495\sin(2\ln\beta_{1022}\cdot 0.6463) \\
&+0.01438\cos(3\ln\beta_{1022}\cdot 0.6463) - 0.03937\sin(3\ln\beta_{1022}\cdot 0.6463) \\
&-0.02212\cos(4\ln\beta_{1022}\cdot 0.6463) + 0.06559\sin(4\ln\beta_{1022}\cdot 0.6463) \\
&+0.0277\cos(5\ln\beta_{1022}\cdot 0.6463) - 0.02724\sin(5\ln\beta_{1022}\cdot 0.6463) \\
&-0.005375\cos(6\ln\beta_{1022}\cdot 0.6463) + 0.009788\sin(6\ln\beta_{1022}\cdot 0.6463) \\
&+0.01021\cos(7\ln\beta_{1022}\cdot 0.6463) - 0.01863\sin(7\ln\beta_{1022}\cdot 0.6463) \\
&-0.01368\cos(8\ln\beta_{1022}\cdot 0.6463) + 0.004736\sin(8\ln\beta_{1022}\cdot 0.6463)
\end{aligned} & \text{if } -13 \leq \ln\beta_{1022} \leq -7.5, \\[4pt]
1.28 & \text{if } \ln\beta_{1022} > -7.5.
\end{cases}
\tag{4}
$$

We summarize different existing parameterization functions for different past time periods in the supplementary materi-
als. (All except the last parameterization therein were used to make the SAGE-4λrecord.) The last table entry there is the
parameterization given above, which was derived from SAGE III data and used for the HTHH-MOC activity. Because
SAGE III has the highest number of useful channels across a broad spectral range out of all satellite data sets, we further





recommend this parameterization for general use, i.e. for time periods not covered by observations, i.e. outside of 1979-
2023.

One example of the 2-wavelength case is the forcing for the HTHH-MOC activity. The observational data set used for
this retrieval was GloSSAC version 2.22, which provides full coverage on 2 wavelengths (525, 1020 nm. With GloSSAC
being short one channel to allow for a *full retrieval*, one free parameter needed to be eliminated. SAGE III data available
between June 2017 and December 2023 provided 6 wavelengths of good quality (out of the 9 total) over the time period
of interest and was therefore used as the multi-$\lambda$ data set (see turquoise box in Figure 1) to get the empirical relationship
between measured AEC and $\sigma$.

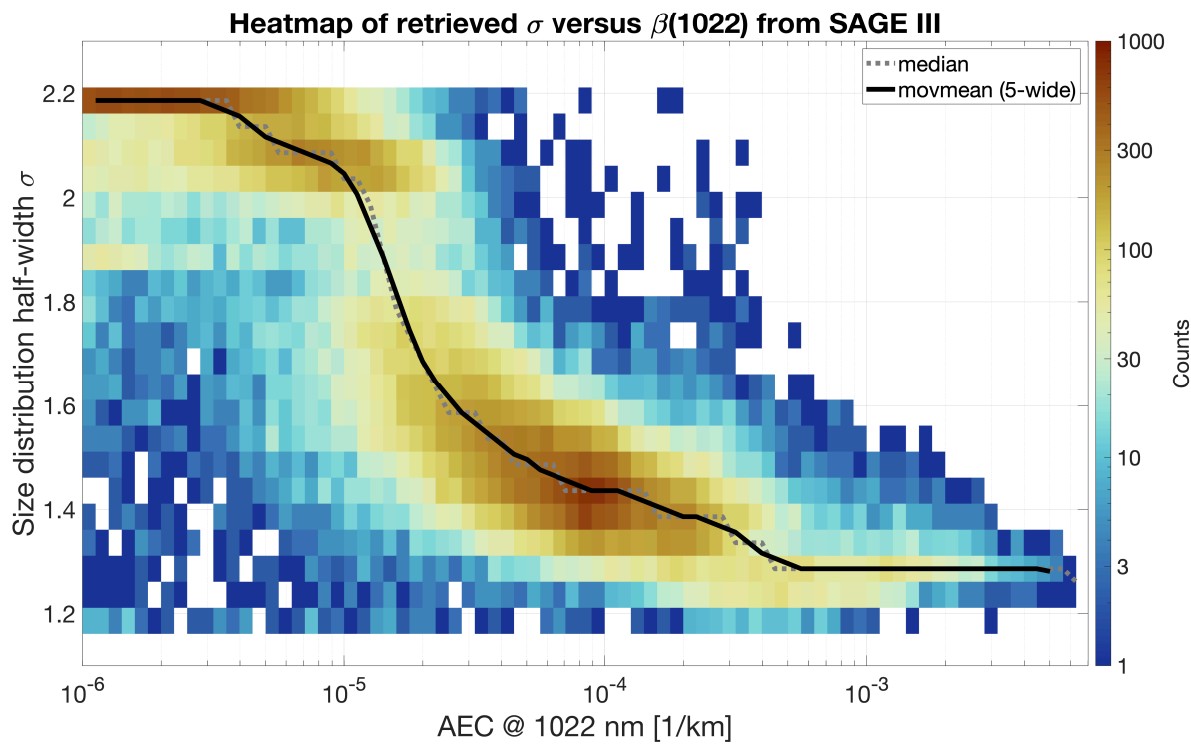

**Figure 3.** 2D histogram of all $\sigma$ values retrieved from data of 6 selected SAGE III wavelengths (June 2017 – December 2023, above 20
km) against their corresponding aerosol extinction coefficients. The dotted grey line shows the median value for every extinction coefficient
bin, the solid black line shows the moving mean of the median at a width of 5 bins. A function can be fitted to the black line to get the
parameterization function $\sigma(\beta_{1022})$.

(c) **Periods with only single-wavelength aerosol extinction measurements.**

Under these conditions, an additional parameterization of $r_m$ or $\sigma$ (whichever has not yet been parameterized) as a





function of a good-quality wavelength replaces yet another free parameter, leaving the aerosol number density $n$ as the only remaining unknown, which is retrieved using the only measured AEC $k_{obs}$. The algorithm finds the value for $n$ that minimizes $D$. For the historic SAGE-3$\lambda$ record, more sophisticated methods were used in this case, which are documented in S1.

## 3.2   Wavelength quality

Assessing the quality of each available wavelength is vital for a consistent and useful size distribution retrieval. The number of good quality wavelengths available determines the protocol for REMAP, as described in the previous section. Satellite measurements rely on some algorithm to retrieve AEC data. Such an algorithm may already make assumptions about the aerosol size distribution. This is necessary to produce data in the first place, however, the procedure must be checked before using satellite data for REMAP. If there are strong constraints on the product, or biases or low accuracy are reported for

any channels of a satellite product, care must be exercised, when selecting REMAP input data. In composite products that synthesize different data sets, measurements are already processed and may have been transposed onto new wavelengths. If this is the case, or if a number of wavelengths have been made into a greater number of wavelengths, the method used must be checked, as the information content in each channel may be limited. Ideally, each channel should represent a separate physical measurement.// Selecting good quality wavelength was essential for the SAGE III parameterization. In this case using all 9

wavelength channels leads to an ambiguous (non-monotonic) parameterization (see Figure S5 in supplements). Only after assessing each wavelength and various wavelength combinations, did we achieve the correlation in Figure 3. This principle is further illustrated in Figure 4, which shows the ratio of calculated divided by observed AEC from one month of SAGE III. The size distribution retrieval was performed on a selection of 6 (black font) out of the 9 available wavelengths. From this the theoretical AEC were again calculated on all measurement wavelengths, as to compare the result on all channels. The 3

wavelengths that were omitted in the retrieval show significantly biased ratios, while the other 6 achieve agreement within $\pm 2.5\%$ in the Junge layer region. The biases are expected, since the measurements on the biased channels did not contribute to the constriction of the size distribution parameters. However, the ratios in the other channels diverge significantly from 1 if the retrieval is performed on all 9 wavelengths. Thus, the retrieval was improved by omission of biased channels. To identify biased channels, ratios could be calculated for many different combinations of retrieval wavelengths and then compared, but

this is a lengthy process. In the next two sections, we describe two analytics that help evaluate the wavelength quality of each channel of an observational data set.

### 3.2.1   Data scatter

Shorter wavelengths tend to be scattered more due to the higher molecule-to-aerosol extinction ratio. (Because of this sensitivity, shorter wavelengths are also more susceptible to uncertainties in pressure and temperature.) This effect becomes most

pronounced for AEC $< 10-4$ and is visible in Figure 5. But also for longer wavelengths, the instrument design and sensitivity can produce systematic data scatter fingerprints. This is exemplified in SAGE III, which is mounted on the ISS, a unique platform with distinct challenges. Leckey et al. (2021) describe these in detail alongside the advantages. A key feature that





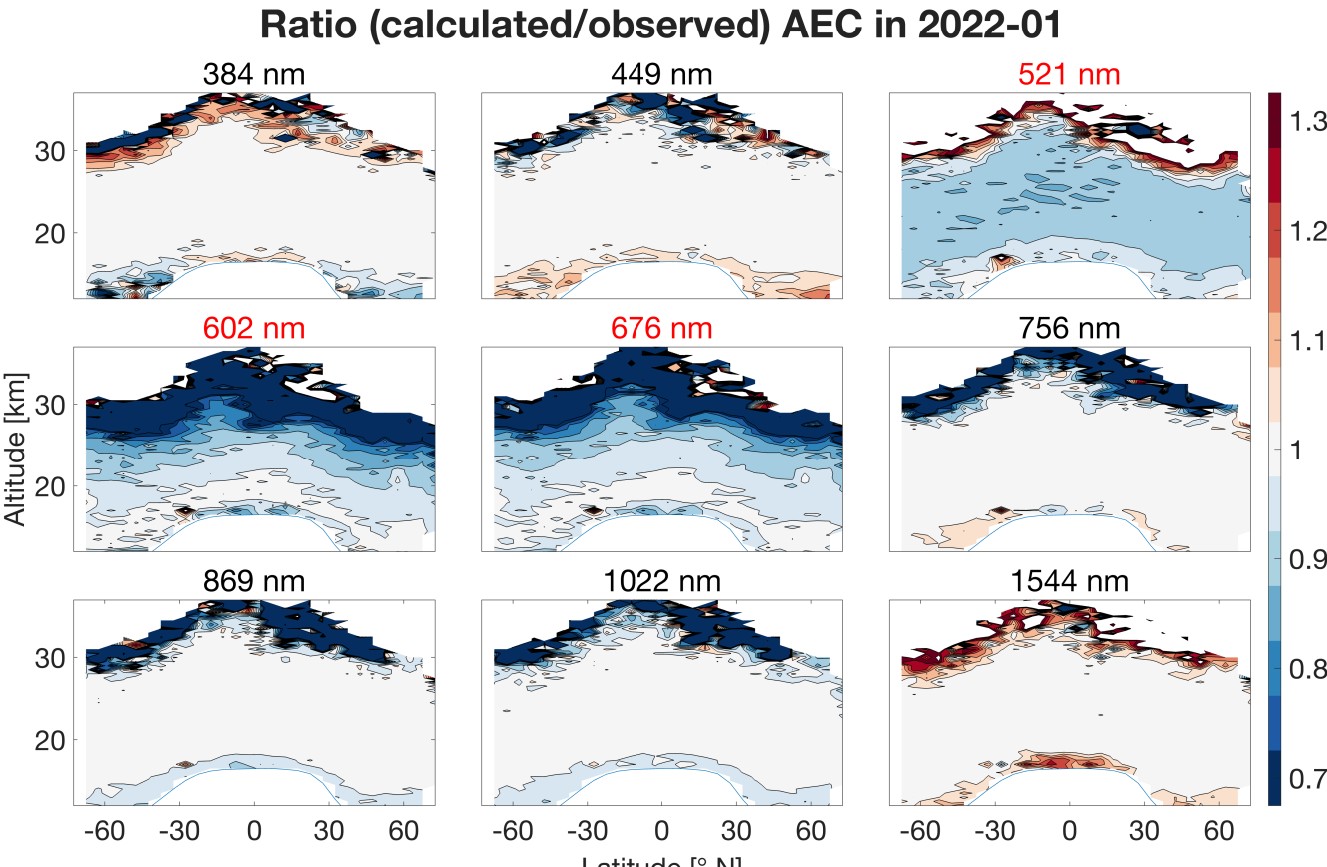

**Figure 4.** Ratios of the calculated and measured monthly mean AEC from SAGE III for January 2022. The calculated AEC are based on a retrieval with 6 good quality SAGE III wavelengths (black font), with three wavelengths (521 nm, 602 nm and 676 nm; red font) removed due to systematic biases. From this retrieval of selected wavelengths, theoretical AEC are again calculated on all 9 original wavelengths to demonstrate the biases.

decreased the signal-to-noise ratio for more than half of the occultation events is an optical window designed to avoid contamination of the instrument as rockets access the ISS. In comparison, SAGE II was flown on a smaller, unserviced spacecraft that did not receive exhaust gases from docking maneuvers altogether. Data scatter is somewhat corrected for in data sets that include standard deviations (e.g. GloSSAC), where the algorithm can then assign less weight to data points with a higher standard deviation. Based on Figure 5 we rejected the wavelengths 602 nm and 676 nm, because the ratios scatter strongly off the 1:1 line even for altitudes below 30 km. The wavelength 521 nm also shows significant bias towards underestimated calculated AEC already at low altitudes. This is consistent with Figure 4, where the 521 nm panel widely shows a low bias around 5 % for the calculated/observed ratio.

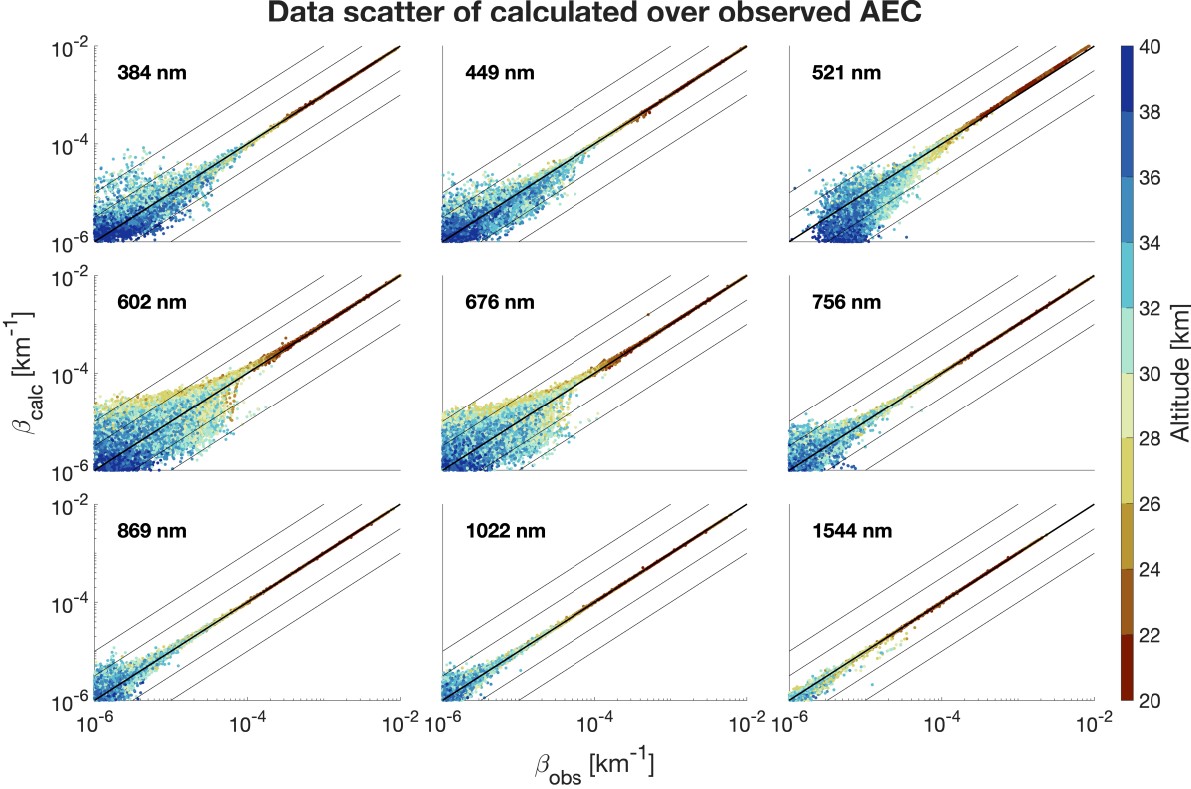

**Figure 5.** Scatter plots correlating calculated and measured AEC $\beta$ on the 9 SAGE III channels for all measurements above 20 km. To calculate the AEC from Mie theory, the SLN size distribution was retrieved by fitting its parameters to all 9 wavelengths simultaneously. The calculated AEC were then computed on all same 9 wavelengths again. Symmetric scatter around the thick black line indicates a poorer signal-to-noise ratio, while the center of the point cloud veering off the line indicates a bias on the affected wavelength. The colors are coded to the height of the measurement, with smaller AEC typically coming from greater heights.

### 3.2.2 Color index

The theoretical AEC given by Mie theory have a wavelength dependence. This dependence can be seen in the log-log color index plots in Figure 6. The blue and red points are the median calculated and measured SAGE III AEC between 7 – 12 km above the mean tropopause. Two *full retrievals* are shown, one with all 9 SAGE III wavelengths and one with exclusion of the three wavelengths in red font in Figure 4. Since the blue points follow from Mie theory, they represent an idealized AEC function of wavelength. The measured, red points do not perfectly trace out such an idealized dependence, if there are biases in any of the wavelengths. This is evident in the left panel. Because REMAP finds the size distribution parameters that best fit the observations, it is not in all cases clear from these color index plots, which wavelengths are biased. However, this analytic definitively shows that the chosen ensemble of 6 wavelengths in the right plot yields theoretical AEC that are in better





agreement with the measured AEC, reducing the root mean square error (RMSE) to a fourth and the mean absolute percentage error by 0.75 %.

## Wavelength dependence of median AEC (5-12 km above tropopause)

**Figure 6.** Color index plot of the calculated and measured SAGE III median aerosol extinction coefficients between 5 and 12 km above the mean tropopause, illustrating their wavelength dependence. The calculated aerosol extinction coefficients on the left are based on a size distribution retrieval using all 9 SAGE III wavelengths, whereas a selection of 6 good quality wavelengths is made for the retrieval for the plot on the right. The blue points are governed by Mie theory and trace out an ideal wavelength dependence, while also trying to fit the observations as best as possible. The residual mean square errors (RMSE) and mean absolute percentage errors (MAPE) both decrease with a retrieval based on selected good quality wavelengths, improving the fit.



### 3.3 Calculation of aerosol properties

In this next step we calculate the AEC, SSA and AF for the wavelength bands of individual global models, using Mie theory. The extinction coefficient $\beta$ of an aerosol ensemble at a given wavelength $\lambda$ is given by

$$\beta(\lambda) = \int \frac{dn(r)}{dr} C_{ext}(\lambda, r) dr, \tag{5}$$

where $C_{ext}(\lambda, r)$ is the extinction cross section of a spherical droplet with radius $r$, which is readily calculated by Mie theory (Bohren and Huffman, 1998). The calculation of $C_{ext}(\lambda, r)$ requires the real and imaginary parts of the refractive index as a function of RH and temperature. We obtain the refractive indices from Luo et al. (1996) for wavelengths between 0.35 µm to 2 µm. For $\lambda < 0.35$ µm, we simply use the refractive index at 0.35 µm. We consider the imaginary part of the refractive index for $\lambda < 2$ µm to be zero, i.e. negligible absorption. This assumption is well justified, because the absorption in the wavelength range of 0.2-2 µm of aqueous sulfuric acid solution is indeed very small (Jonasz and Fournier, 2007). For $\lambda < 0.2$ µm, there are some strong absorption features for aqueous $H_2SO_4$-$H_2O$ solutions, however the power in the solar radiation for $\lambda < 0.2$ µm is negligible compared to the total solar radiation. For wavelengths longer than 2 µm, we use the values reported in Biermann et al. (2000), which extend both the real and imaginary part up to 20 µm. Beyond that point the refractive indices remain constant at the last reported value. In Eq. (5), the term $dn/dr$ characterizes the SLN size distribution (Eq. (1)) that the extinction cross section is multiplied with along the radius dimension and then integrated.

As the global models operate with wavelength bands, we need to perform a weighting of $\beta(\lambda)$ over a wavelength region assuming Planck's law for a black body:

$$B(\lambda, T_b) = \frac{2hc^2}{\lambda^5} \frac{1}{exp\left(\dfrac{hc}{\lambda k_B T_b}\right) - 1}. \tag{6}$$

Here, $B$ is the radiation intensity, $h$ is the Planck constant, $c$ the speed of light, $k_B$ the Boltzmann constant, and $T_b$ is the temperature of the black body. A black body temperature of 5900 K is used for the solar bands and 255 K for the terrestrial bands (representing the radiative temperature of the planet Earth). The weighted extinction coefficient $\overline{\beta}$ for the band from wavelength $\lambda_1$ to $\lambda_2$ is calculated as

$$\overline{\beta}(\lambda_1, \lambda_2) = \frac{\int_{\lambda_1}^{\lambda_2} B(\lambda, T_b)\beta(\lambda)d\lambda}{\int_{\lambda_1}^{\lambda_2} B(\lambda, T_b)d\lambda} \tag{7}$$

The single-scattering albedo $\omega$ is the ratio of weighted scattering coefficient $\overline{\delta}$ to weighted extinction coefficient

$$\omega = \frac{\overline{\delta}}{\overline{\beta}}, \tag{8}$$





where $\overline{\delta}(\lambda_1, \lambda_2)$ for the band bounded by $\lambda_1$ to $\lambda_2$ is calculated analogously to $\overline{\beta}$ from Eqs. $(4 - 6)$.

The asymmetry factor $g$ for a given wavelength $\lambda$ of a droplet of radius $r$ is defined as (Bohren and Huffman, 1998):

$$g(\lambda,r) = \frac{\int_\Omega \frac{dC_{sca}(\lambda,r,\theta)}{d\Omega} cos(\theta) d\Omega}{C_{sca}(\lambda,r)} \tag{9}$$

with scattering cross section $C_{sca}(\lambda,r)$ and differential scattering cross section $dC_{sca}/d\Omega$. The asymmetry factor for a given wavelength band $g(\lambda_1, \lambda_2)$ is given by integrating over the size of particles and over all wavelengths between the lower and upper band limits $\lambda_1$ and $\lambda_2$:

$$g(\lambda1, \lambda2) = \int\limits_{\lambda1}^{\lambda2} \int\limits_{r} g(\lambda,r) C_{sca}(\lambda,r) \frac{dn(r)}{dr} B(\lambda,T_b) dr d\lambda \frac{1}{\overline{\delta}(\lambda1,\lambda2) \int_{\lambda1}^{\lambda2} B(\lambda,T_b) d\lambda}. \tag{10}$$

## 3.4  Products for global climate models

For the treatment of heterogeneous chemistry on the surfaces of stratospheric aerosol particles, primarily the SAD is required (e.g. for the ubiquitous $N_2O_5$ hydrolysis). In addition, for some reactions occurring in the bulk of the particles, the mean radius $r_{mean}$ is required, e.g. for HOCl + HCl, which is important under cold conditions, see Hanson et al. (1994). For the treatment of radiative effects of aerosols, the state-of-the-art CCMs have their own radiation schemes, which typically rely on three optical quantities for all model-specific wavelength bands: AEC, SSA and AF (see Figure 1).

### 3.4.1  Mass-related quantities

CCMs that explicitly treat heterogeneous reactions require the aerosol SAD to describe the kinetics. The rate coefficient of a heterogeneous reaction on aerosol surface R is given by

$$R = \frac{\gamma \, \text{SAD} \, \overline{v}}{4}. \tag{11}$$

Here, $\gamma$ is the reactive uptake coefficient and $\overline{v}$ the mean thermal velocity. The radius dependence of $\gamma$ was explored by Hanson et al. (1994), providing a framework for the reactive and diffusive properties of gases accommodated on liquid droplets. The mean radius $r_{mean}$ and effective radius $r_{eff}$, as well as the SAD, aerosol volume density $V$ and $H_2SO_4$ mass $m_{H_2SO_4}$ can be calculated from the number density $n$, the mode radius $r_m$ and the half-width $\sigma$ of the retrieved SLN size distributions. For the mass of sulfuric acid the mass fraction $w(T)$ and aerosol density (computed as a function of temperature) $\rho(T)$ are additionally required:

$$r_{mean} = r_m \exp\left[0.5 \ln^2 \sigma\right] \tag{12}$$





$$\mathrm{SAD} = n4\pi r_m^2 \exp\left[2\ln^2\sigma\right] \tag{13}$$

$$V = n\frac{4}{3}\pi r_m^3 \exp\left[4.5\ln^2\sigma\right] \tag{14}$$

$$m_{\mathrm{H_2SO_4}} = V\rho(T)\,w \tag{15}$$

360  $$r_{eff} = \frac{3V}{A} = r_m e^{2.5(\ln\sigma)^2} \tag{16}$$

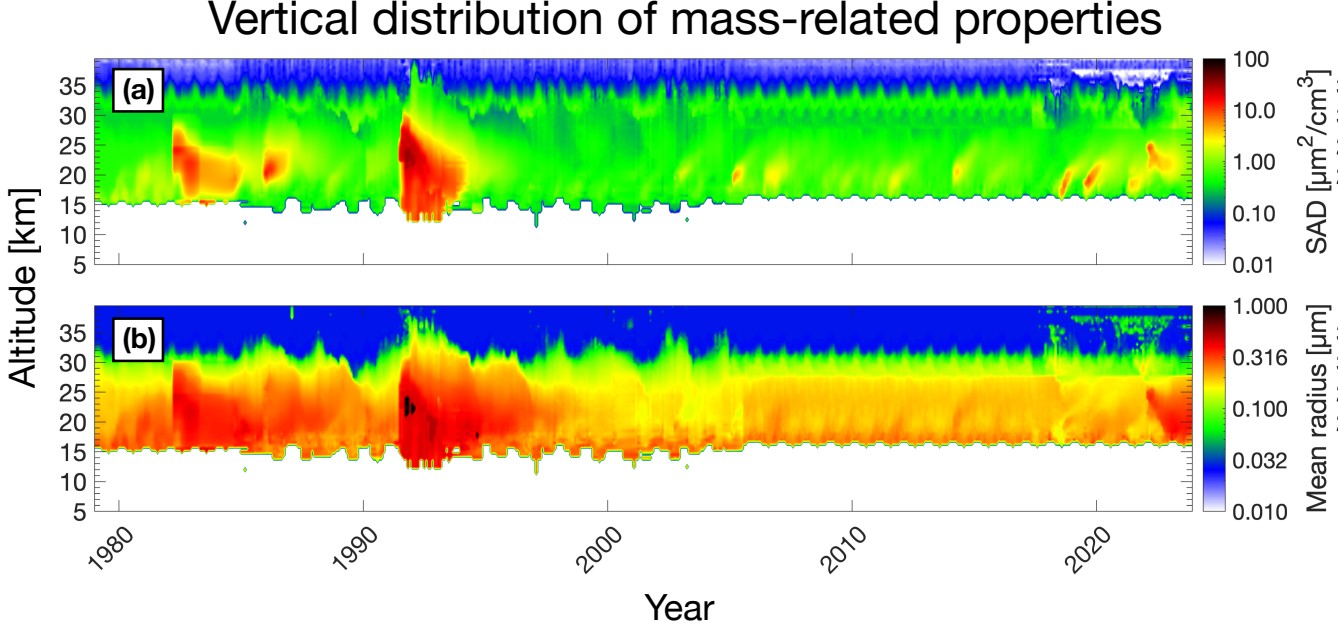

**Figure 7.** Surface area density (SAD) and mean radius for the time period 1979 and 2023 derived from the REMAP-GloSSAC-2023 record for the latitude band $0-5°$ N.

The values of SAD and $r_{mean}$ are shown in Figure 7 for the latitude band $0-5°$ N. During volcanically quiescent times, the stratospheric SAD is $0.5-1\,\mu\mathrm{m}^2\mathrm{cm}^{-3}$. After large volcanic eruptions the SAD can reach $20-40\,\mu\mathrm{m}^2\mathrm{cm}^{-3}$.



### 3.4.2 Optical properties

In GCMs and CCMs, the optical properties AEC($\lambda$), SSA($\lambda$) and AF($\lambda$) are required for each wavelength band to compute the
radiative forcing exerted by the aerosol on the model. Some models need only the sulfate mass $m_{H_2SO_4}$ and from there estimate
the radiation effects; therefore, for these models we provided the total mass density of $H_2SO_4$ or other mass-related quantities
as function of latitude, altitude and time.

We provided these radiative properties for the CCMI-1, CMIP6, CCMI-2022 and HTHH-MOC models listed in Table 1 and
they are openly available as SAGE-3$\lambda$ (Luo, 2013), SAGE-4$\lambda$ (Luo, 2017), REMAP-CCMI-2022-ref (Luo, 2020), REMAP-
CCMI-2022-sai (Jörimann, 2023) and REMAP-GloSSAC-2023 (Jörimann, 2024) data set (see "Data availability").

Figure 8 shows examples of AEC($\lambda$), SSA($\lambda$) and AF($\lambda$) for two wavelength bands of the GCM ECHAM6. The upper panels
(a-c) show the results for one solar band, 0.442 – 0.625 µm, the lower panels (d-f) for one terrestrial infrared band, 10.204 –
12.195 µm. The SSA in the visible band is unity by definition (assuming no absorption, i.e. only the real part of the refractive
index matters). The backscattered solar radiation reduces the power of the incoming solar radiation, exerting an overall cooling.
However, for the infrared band, 10 – 12 µm, the SSA is nearly zero. Then the largest part of the extinction is due to absorption,
which is why the stratospheric aerosol leads to in situ heating of the lower stratosphere.

### 3.5 Method using model output

For the CCMI-2022 sensitivity experiment senD2-sai, an aerosol forcing simulating man-made stratospheric aerosol injection
(SAI) needed to be uniformly prescribed in different models (Plummer et al., 2021). The forcing was derived from a single-
model ensemble run in the Whole Atmosphere Community Climate Model, version 2 of the Community Earth System Model
(WACCM-CESM2). To apply REMAP to model output – instead of observations – the workflow was altered.

Instead of retrieving size distribution information, the output variables aerosol mode wet diameter and number density of the
model's trimodal distribution were directly used for the Mie theory calculations. Relative humidity, temperature and pressure
fields were also readily available from the same source (cf. Table 1). For the three aerosol modes represented in WACCM-
CESM2, we assumed a half-width of $\sigma = 1.6$, $\sigma = 1.6$ and $\sigma = 1.2$ respectively. With this data, the AEC and the SSA are readily
calculated for any spectral band using Eqs. (5 – 7). The AF is given by Eqs. (9) and (10), while the mean radius and SAD are
calculated with Eq. (12) and Eq. (13), respectively.

### 4 Validation

Recently Trickl et al. (2023) published results of 45 years of ground based LIDAR measurement of stratospheric aerosols since
1976 at the station Garmisch-Partenkirchen (47.5° N, 11.0° E, Southern Germany). The integrated backscatter coefficients –







**Figure 8.** Aerosol radiative properties derived from REMAP-GloSSAC-2023: aerosol optical depth (AOD) above 10 km altitude, single-scattering albedo (SSA) and asymmetry factor (AF) at 20 km altitude from 1979 to 2023. (a – c) For one of the solar bands, 0.442 – 0.625 µm of the ECHAM6 GCM. Panel (b) has constant data, because all aerosol extinction is due to scattering, for visible light there is no absorption at 20 km. (d – f) For one of the terrestrial bands, 10.204 – 12.195 µm.

a quantity required for SSA – at 694.3 nm can also be calculated using the REMAP method. The agreement between the



measured (red symbols and line in Figure 9) and the calculated integrated backscatter coefficients (black line in Figure 9) is very good. It should be noted that here we are comparing the zonal mean value of a 5° latitude band created with REMAP to the more fluctuating values from a single station. REMAP achieves excellent agreement after large volcanic eruptions (e.g. El Chichón in 1982, Pinatubo in 1991 and Raikoke in 2020), but slightly underestimates the sustained volcanic burden in the episode of moderate volcanic eruptions around 2010. The retrieved integrated backscatter is also consistently high-biased

in the background state (visible between 1997 and 2004), although other stations also report higher values than Garmisch-Partenkirchen in these years (Trickl, 2024; Personal communication). We can trace this overestimation back to the lowest levels in the stratosphere, where the single-mode log-normal distribution assumption does not hold and non-sulfuric aerosol is sometimes present, which cannot be correctly interpreted in REMAP. Especially large deviations accordingly occur during large wildfires without any concurrent volcanic eruption, e.g. during and after the British Columbia wildfires (BC fires in Fig-

ure 9) 2018. In this comparison, even for thes poorly captured events, the errors stay well within a factor of 2.

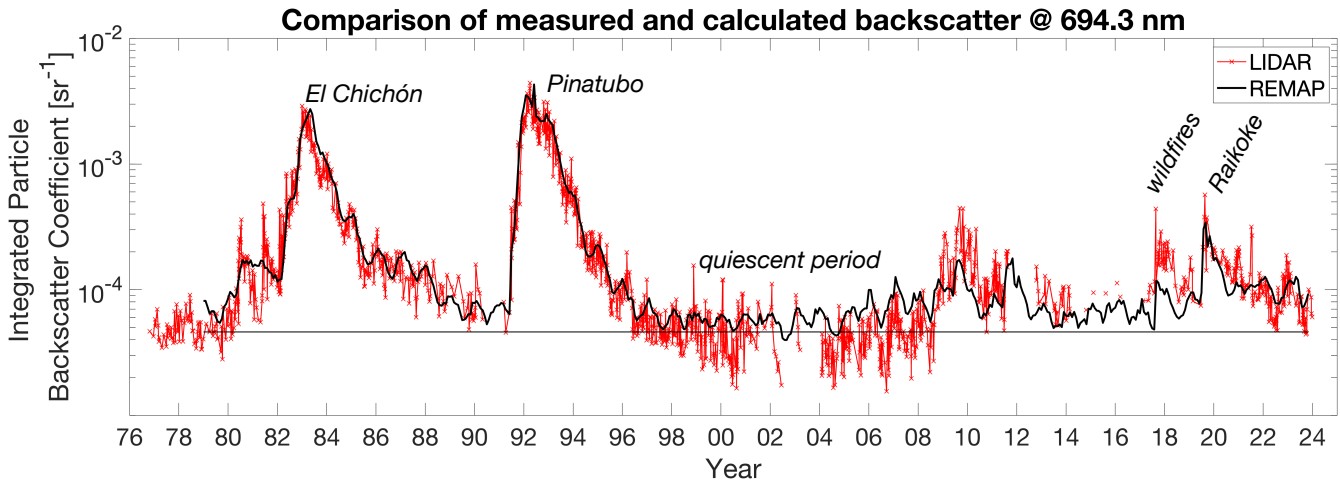

**Figure 9.** Integrated backscatter coefficients of stratospheric aerosol (integrated from 1 km above the tropopause to the upper end of the layer) over Garmisch-Partenkirchen, Germany (47.48° N, 11.06° E, 734 m asl). Red symbols and line: measured by the ground based LIDAR at 694.3 nm. Several significant events are marked: the volcanic eruptions of El Chichón 1982, Pinatubo 1991 and Raikoke 2020, an extended quiescent period in 1997-2004 and the British Columbia (BC) wildfires. From Trickl et al. (2023). Black line: zonal average monthly mean integrated backscatter coefficients for the latitude band 45° N to 50° N calculated from GloSSACv2.22 using REMAP with the parameterization $\sigma(\beta_{1022})$ (Eq. (4)).

Since 1971, in situ particle measurements using the optical particle counter, OPC (Rosen and Kjome, 1991), combined with a condensation nuclei counter (CNC) (Delene and Deshler, 2000), were performed on average twice a month at Laramie, Wyoming (41.3° N, 105.5° W, USA). The OPC measures the aerosol number density with radii $r \gtrsim 0.15$ $\mu$m in several size

channels. The CNC detects number density condensation nuclei with radii $r \gtrsim 10$ nm. Uni- or bimodal log-normal aerosol size





distributions have been obtained by fitting the measured counts of individual size channels of the OPC and CNC (Deshler et al.,
2019). Quaglia et al. (2023) then calculated the effective radius as the ratio of the third and second moment of the obtained
size distributions. Panel (a) of Figure 10 shows mean aerosol effective radius after the Mount Pinatubo eruption (June 1991)
between 14 and 30 km height above Laramie. Balloon measurements with one standard deviation are shown in blue and the

corresponding REMAP values (37.5 – 42.5° N) in yellow. The momentary state of the stratosphere, measured by the balloons,
fluctuates more than the monthly mean values captured by SAGE II, which the retrieval is mostly based on at this time. Re-
gardless, the retrieved values mostly stay well within the measurement uncertainty range.

In panel (b) of Figure 10 measured and retrieved vertical profiles of aerosol SAD after the volcanic eruption of Hunga

Tonga-Hunga Ha'apai (January 2022) are compared. The measured profiles are the result of the Balloon Baseline Stratospheric
Aerosol Profiles (B$^2$SAP) campaign (Todt et al., 2023) and were taken at La Réunion (21° S, 55° E, FRA) with Portable
Optical Particle Spectrometers (POPS) roughly a year after the eruption. The volcanic plume at this time is well mixed zonally
and visible as elevated SAD between 20-25 km in both products. REMAP tends to underestimate SAD above 20 km and
overestimate it below, effectively resulting in a slightly shifted peak. The REMAP maxima are also around 10 % smaller than

the balloon maxima. The profile 08-12-2022 also illustrates that REMAP cannot capture tropospheric aerosol (below ~15 km
here), as no satellite extinction coefficient data is available (dotted orange line stays constant). Extinction coefficients were
also calculated for the POPS measurements (dotted blue line), assuming a uni- or bimodal distribution. It is evident that the
retrieved SAD follows the GloSSACv2.22 AEC and the balloon AEC follows the measured SAD. This is logical, as in both
cases the latter is constructed from the former. Therefore the differences between retrieved and measured SAD are mainly

already encoded in the data set used for the retrieval, i.e. GloSSACv2.22. Arguably, the main contributor to these differences
is, again, that satellites provide zonal monthly means, while POPS records the momentary state of the stratosphere locally, with
all the associated variability. A comparison without this limitation would require more abundant in situ data of stratospheric
aerosol.

## 5   Conclusions

The REMAP method is a useful tool for creating stratospheric aerosol forcings required as inpput for climate models and
converting between different aerosol properties. The retrieval algorithm performs well under typical stratospheric conditions,
except close to the tropopause and in regions with very low signal, typically close to the upper edge of the Junge layer. It
has been used for studies on volcanic aerosol in the stratosphere and, more generally, to equip GCMs and CCMs in climate
studies. The main limitation of the retrieval is the underlying assumption of a single-mode log-normal size distribution, which

is necessary for REMAP to flexibly work on many different data sets with limited wavelength channels. Despite this limitation,
REMAP also reproduces the particle size distributions after volcanic eruptions reasonably well. The Mie calculation is based
on theoretical understanding and produces optical properties for each wavelength as well as for each wavelength band. The
output parameter space and its resolution can be costumized at the cost of computational time.



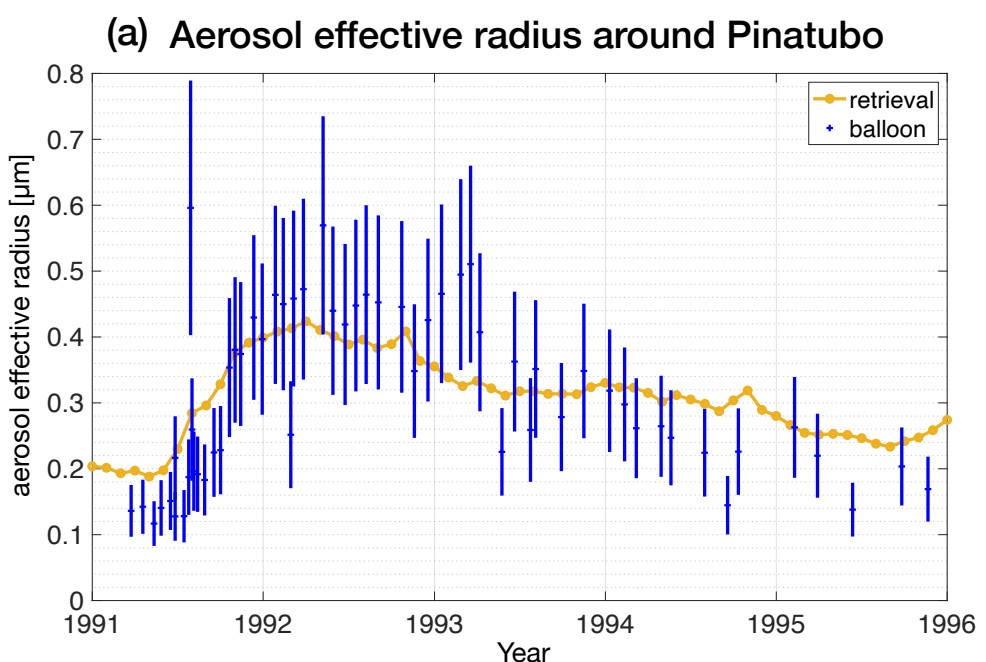

**Figure 10.** Comparison of balloon-borne measurements and REMAP monthly mean values retrieved using GloSSAC. Panel (a) shows mean aerosol effective radius over Laramie WY during the eruption of Mt. Pinatubo in 1991, while panel (b) shows two vertical surface area density and AEC profiles over La Réunion after the Hunga Tonga Hunga Ha'apai eruption in January 2022 . Balloon extinction is not measured, but rather derived after fitting a size distribution to the OPC measurements, whereas retrieval extinction is obtained using REMAP on SAGE III data.





Backbone of all the stratospheric aerosol data sets created with REMAP for different MIPs are the SAGE II and III AEC at
3 and 6 wavelengths respectively, which have provided the most complete coverage over multiple decades. These data are the
basis for deriving important parameterizations that enable retrieval using only two, or even one single channel, thus permitting
the use of the GloSSAC-composite. Between the end of the SAGE II and the beginning of the SAGE III operational period,
there is a gap of over 10 years, meaning that no parameterization of similar quality can be established for this time. While
OMPS-LP data is available starting in 2012, we have not used it with REMAP, because of plans to integrate it into future
GloSSAC versions (Kovilakam et al., 2024). Also, the limited accuracy of the OMPS(NASA) product reported by Taha et al.
(2021) raises the question, whether sufficient quality wavelength channels would be available for a retrieval. As of yet, no other
OMPS-LP product suited for use with REMAP exists. Therefore, we suggest using the SAGE III parameterization to bridge
the gap, which we also used for the entire GloSSAC period that is validated here.

The REMAP AEC calculated from the retrieved size distributions agree well with the measured data. This holds not only in the
visible and near infrared (SAGE wavelengths at 525 (532 in SAGE III) nm and 1020 (1022) nm), but also in the far infrared
(HALOE at 3.46 μm and ISAMS at 12.6 μm; see Figure S3 in the supplementary to this paper for 3.46 μm and Figure S6 of
Arfeuille et al. (2013) for 12.6 μm). We conclude that REMAP realistically reproduces the radiative properties of the strato-
spheric aerosol. The main caveat is that input data must be carefully selected and screened, as biases of individual wavelengths
and contaminated data points can significantly change the quality of the product.


*Code and data availability.* The REMAP code and its products are freely available for download from the ETH research collection:

**REMAPv1**: Jörimann (2025), https://doi.org/10.3929/ethz-b-000715168

**SAGE-4λv2**: Luo (2013), https://doi.org/10.3929/ethz-b-000714581

**SAGE-3λv4**: Luo (2017), https://doi.org/10.3929/ethz-b-000715155

**REMAP-CCMI-2022-ref**: Luo (2020), https://doi.org/10.3929/ethz-b-000715176

**REMAP-CCMI-2022-sai**: Jörimann (2023), https://doi.org/10.3929/ethz-b-000714654

**REMAP-GloSSAC-2023**: Jörimann (2024), https://doi.org/10.3929/ethz-b-000713396

*Author contributions.* BL conceived the method with TP, wrote the REMAP code and produced and analyzed output data sets. AJ extended
and edited the code, produced and analyzed records and wrote the paper. TS contributed to the analysis and writing process and supervised
AJ. GM, GC and TP provided advice and insights for the writing process and edited the manuscript.

*Competing interests.* At least one of the (co-)authors is a member of the editorial board of the Geoscientific Model Development journal.
The authors have no other competing interests to declare.



*Acknowledgements.* This work initiated from CCMI, the Chemistry-Climate Model Initiative of IGAC (https://blogs.reading.ac.uk/ccmi/).
We thank the CCMI leadership for their support and guidance. Andrin Jörimann acknowledges the support from ETH Prof. Louise Harra, APARC (http://www.aparc-climate.org), and the Swiss National Science Foundation (SNSF) project AEON (grant no. 200020E_219166). Timofei Sukhodolov acknowledges the support from the Karbacher Fonds, Graubünden, and the Simons Foundation (SFI-MPS-SRM-00005208). Gabriel Chiodo acknowledges funding from the European Commission via the ERC Starting Grant 101078127. We acknowledge funding from the UK National Centre for Atmospheric Science (NCAS) for Graham Mann via the NERC multi-centre Long-Term Science
programme on the North Atlantic climate system (ACSIS, NERC grant NE/N018001/1. Calculations have been performed at the ETH cluster EULER.



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
