# Peer review of "A REtrieval Method for optical and physical Aerosol Properties in the stratosphere (REMAPv1)"

_EGUsphere, 2025_

## Author Response (AR1)

**Author response**

**Reviewer 1:**

In this paper a method to retrieve stratospheric aerosol properties from observational or model data, named REtrieval Method for Aerosol Properties (REMAP), is described. It is intended especially for use in climate models and model intercomparison projects since the parameters that are most useful for the calculation of the stratospheric aerosol's radiative forcing such as surface area density (SAD), single-scattering albedo (SSA), asymmetry factor (AF) and the aerosol extinction coefficient (AEC) at any desired wavelength are provided through it. All of the mentioned parameters can be calculated relatively simply from a single lognormal (SLN) size distribution, the retrieval of which is the basis of this method. The paper equips the potential user of the method with a useful generalized scheme to retrieve the SLN distribution (and derived quantities) based on the quality and number of available AEC channels. For this a kind of lookup table of AECs calculated by Mie theory for all combinations of the three parameters describing the SLN (median radius, geometric standard deviation and number density) within physically reasonable value ranges is compared to the AECs of the observational or model data set in a best-fit approach. The data retrieved with REMAP are validated with a long-time LIDAR data set and with in situ balloon-borne measurements after the Hunga Tonga – Hunga Ha'apai eruption. Supplementary material is provided on data sets previously generated using the main method described in the paper, on different satellite data sets and corresponding data quality that they are based on and on data gap filling.

The paper is structured logically and both the science and its presentation are of high quality. The abstract and title characterize the paper well. All methods presented and used are valid. The main method is explained well, in part through a well crafted color-coded overview scheme that makes it easy to follow. The language is concise overall, but some imprecise scientific nomenclature is used. All information that is needed to reproduce the results and implement the method is provided in the paper. The appropriate literature is cited throughout the document. There are some smaller errors scattered throughout the document and mathematical formulae that have to be fixed. Further context should be added in some places.

The paper makes a valuable contribution to modelling science, since it provides climate modelers with a path towards a best guess of the relevant properties for stratospheric aerosol radiative forcing, which is important for climate modelling.

Therefore, I recommend the paper for publication after minor revisions.

**Response:**

We thank the reviewer for taking the time to provide valuable and detailed specific comments and technical corrections. We adopt all corrections that are not further mentioned and address the rest, as well as all other comments below one by one.

Line 62: The absorption of terrestrial radiation that is also calculated by Mie theory in the process of calculating the aerosol forcing should be mentioned in this sentence.

We agree with the reviewer and changed the sentence accordingly.

Line 87: In addition to the mention of Thomason et al. (2008) and maybe even more fitting here would be a citation of Knepp et al. (2024), who show that there is not enough information within SAGE extinction spectra to accurately retrieve bimodal lognormal distributions.

We also changed this sentence to include the new reference.
*Reference added*:

Knepp, T. N., Kovilakam, M., Thomason, L., and Miller, S. J.: Characterization of stratospheric particle size distribution uncertainties using SAGE II and SAGE III/ISS extinction spectra, Atmospheric Measurement Techniques, 17, 2025–2054, https://doi.org/10.5194/amt-17-2025-2024, 2024.

Line 110: You should also mention here in some way, that in order to retrieve AECs from Limb scatter measurements (i.e. from instruments like OMPS-LP, SCIAMACHY, OSIRIS...) the aerosol particle size distribution has to be assumed beforehand. Since you want to learn about aerosol size from the AECs, this is an important, unknown and mostly unavoidable source of error when applying your method to limb scatter measurements.

We mention this complication later in the text (in section 3.2 Wavelength quality). We add one sentence to be more specific about limb scatter measurements.

Equation (1): The logarithm outside of the parenthesis (in the denominator of the fraction) should not be squared. See equation (38) in Grainger (2023). For reference, your σ is "S" in this document.

Agreed and fixed.

Throughout the document: The parameter rm that appears in your equation (1) is the "median radius". The "mode radius" describes the peak location of the size distribution, but since the SLN distribution is asymmetric (in linear radius space) this is not equal to the median radius. Please change "mode radius" to "median radius" throughout the document. Similarly the name "half-width" for σ is not quite correct here, please use "geometric standard deviation" instead.

We agree that this is more precise and using mode radius could lead to confusion. We performed calculations in log-space, which is likely why "mode radius" was previously chosen. To be consistent with literature we updated all the names to "median radius" and "geometric standard deviation", respectively.

Sect. 3.1 (a): In the 3-or-more-wavelengths case just the single solution with the lowest difference score D is taken as the solution, correct? There are probably often other solutions with almost as low difference scores. Are the parameter values of these solutions similar or do they scatter strongly?

This is an important point; when we let all three parameters of the size distribution range freely, the solutions can scatter strongly. This is why we still parameterize one parameter, which leads to less scatter close to the solution. The parameterization itself does not suffer from the "noise", because it is derived from median values.

Line 201 and equation (2): I think it is not correct to call the parameter ς the "standard deviation" as you also talk about the standard deviation of individual data points in line 291 (which does not make sense). I think what you mean is the measurement uncertainty or uncertainty of the data point. In case I am correct please change it to something similar throughout the document.

We understand that the description of the parameter ς was insufficient and change it to "zonal standard deviation", which is what is given in the GloSSAC data set and is what we use here.

Line 213 and Figure 2: Here you write that the lowest values of E in Figure 2 are in the core of the Junge layer. But in Figure 2 the lowest values are at around 25 km altitude, and the "core" of the Junge layer is typically more around 20 km and below, depending on which parameter you look at and depending on latitude. So this sentence should be changed. Also, why is this area of lowest errors at such high altitudes?

This is indeed a wrong description. We change the paragraph and reason that the error generally decreases at higher altitudes (up to the climatology), because of the onion peeling method behind occultation retrieval. This technique is used to derive vertical profiles of atmospheric constituents from observations, i.e. "peeling back" the layers of the atmosphere to determine the particles at different altitudes, making the results for lower altitudes depend on the results on higher altitudes..

Line 215-216: Here, you give a reason for the higher errors above 27 km. Please also mention the reasons for the higher errors in the lowest stratospheric region, which you mention in the same sentence.

We change the paragraph as mentioned above to include this as well. The occultation measurement performs worst in this region and tropospheric intrusions can further decrease the quality of the retrieval.

Line 221-223: These sentences are worded a bit confusing to me. For the Mie calculation only the refractive index (and size distribution) should be needed. RH and Temperatures are needed to calculate the refractive index, not directly for the Mie calculation. Please restructure the sentences to make this clearer.

This is right, we change the text accordingly.

Section 3.1 (c): Taking both rm and σ from parametrization and only retrieving the number density from a single wavelength would likely lead to very unreliable data. An error in rm or σ individually should already have a potentially strong effect on the retrieved number density, since larger particles scatter much more strongly than smaller particles in this size regime. E.g. a high-bias in rm should lead to a low bias in the number density. Now both other parameters of the SLN distribution are parametrized in addition to likely errors introduced by the SLN assumption itself, which also usually affects the number density the strongest. These errors would of course also propagate to the other quantities you calculate from the size distribution, especially the SAD.
Therefore, it should be at least shortly discussed that this retrieval product based on only one wavelength is attached to a lot of uncertainty and likely more or less unreliable (although it might still be the best possible guess for those periods).

We agree with the reviewer, this is exactly that - our best guess for a very limiting scenario. We add a sentence to point out the potential poor quality of the method for this scenario in the text.

Figure 7: To my knowledge, during the Pinatubo period GloSSAC is based mostly on SAGE II, which had big data gaps in the lowermost Junge layer due to the opacity of the lower tangent heights, but you have continuous retrieval data down to 15 km here. Which measurements are these retrievals based on in this case?

The GloSSAC product (at least in recent versions) bridges the gaps in SAGE II observations with a variety of other data sources: a lot of data are linearly interpolated from points within 2 months, others are empirically scaled (to SAGE II wavelengths) from CLAES and HALOE. There is also ASAP-based tropical LIDAR data and other ways to fill gaps. While these methods fill the gaps, they are likely to come with large errors. We do not discuss this in the text, since we purposefully use the final GloSSAC product to avoid data treatment discussion (except in the supplementary for the SAGE-3lambda record).

Figure 9: I am surprised about the good agreement for the Pinatubo period due to a few layers of issues: Firstly, there is the SAGE II data gap in the lowermost Junge layer mentioned in the previous comment, which would likely reduce the amount and quality of observational AEC channels available for REMAP. Secondly, you parametrized σ here, effectively only retrieving two of the three SLN distribution parameters. And last but not least, the Pinatubo eruption led to a clear bimodal particle size distribution

(as we know from OPC measurements) and you can only retrieve a SLN distribution, which would also be a big source of error. Possibly, the second mode with larger particles dominated the measurement signal so strongly, that the SLN retrieval closely approximates only this second mode and therefore the main scattering signal with the first mode with smaller particles being negligible in terms of scattered radiation?
This is not really a clear question, but if you have thoughts on this, maybe you can specify them in the paper.

We agree that the agreement is surprisingly good and we would have still been satisfied if the error had been greater. For one, it is worth mentioning that the observational data here comes from one single station, where - perhaps - the conditions were especially favourable for the REMAP retrieval method. Sigma is in fact parameterized, the reason being that this has consistently produced better results, even though the parameterization is done on a time period that does not include the Pinatubo event itself. This was another surprising result, but one that appears to confirm the robustness of the parameterization. In order to address the bimodal nature of the size distribution after the eruption that is rightly pointed out by the reviewer: we also wondered why this did not introduce larger errors and therefore conducted the test on how REMAP handles unimodal and bimodal representations of the same aerosol. We report this test in the supplementary materials, where we show that the extinction coefficients (and by extension all the optical properties) are very well reproduced in both cases. It appears that the retrieval is flexible enough to find a size distribution that accurately reproduces the optical properties of the associated aerosol. We want to emphasize at this point that this does not necessarily guarantee that the geometrical variables like median radius or surface area density are also of this same high quality.

Line 426: You state that there is no satellite extinction coefficient data below ~ 15 km in the 08-12-2022 REMAP profile and that therefore the dotted orange line stays constant. However, It does not stay constant but does show noise. Where is the information coming from here?

We thank the reviewer for pointing out this inconsistency. This is attributable to a mistake in data analysis and has been corrected. An explanation is given in the *dedicated section at the end of this document*.

Line 430-432: You state that the main contributor to the differences between in situ and REMAP data in Figure 10 (b) is arguably that zonal monthly means of REMAP are compared to the momentary local measurements of POPS. I am not sure if this should be assumed to be the explanation for the substantial difference in altitude of the signal peaks. For this to be the case the plume of Hunga Tonga would have to have been at much lower altitudes at other locations or times within the depicted months, which I don't know to be the case, especially since this difference is so similar at these two different points in time. The much higher size resolution of POPS as opposed to the SLN retrieval with REMAP could just as well be an important if not dominating factor here. A second mode in the true aerosol particle size distribution would (following von Savigny and Hoffman (2020)) likely lead to an underestimation of the REMAP number density as the AEC spectrum would be more strongly affected by the second mode with larger particles, but it would be captured well by POPS, which could possibly explain the differences in SAD seen in Figure 10 (b).

*We address this in the dedicated section at the end of this document.*

Line 458-459: Additionally to biases of individual wavelengths and contaminated data points the quality of the product can also strongly depend on the measurement geometry, e.g. solar occultation vs limb scatter measurements, as for the latter AECs can only be retrieved after assuming the particle size distribution.

We add another sentence here that reiterates this point that is initially made in the chapter about wavelength quality.

Throughout the document: In the whole document the name "SAGE III" is used when "SAGE III/ISS" is meant. However, there also was another SAGE III instrument: SAGE III/M3M, that operated between 2002 and 2005 on the Meteor-3M satellite. Please change "SAGE III" to "SAGE III/ISS" in all places, where you don't write generally about the SAGE III instrument type, but specifically about the SAGE III/ISS instrument.
Also, as a suggestion for future use of the algorithm: SAGE III/M3M should be a very beneficial data set for the use of REMAP both for the retrieval itself and for parametrization, since it covers the same number of aerosol channels as SAGE III/ISS and with overall similar quality. It covers different latitudes however, which may be both an advantage and a disadvantage.

We added a disambiguation at the first mention of SAGE III. We agree that there is an opportunity of using additional data sets in the future.

In the supplementary, in S4: Here you argue that the monomodal retrieval can reproduce the aerosol extinction coefficient (AEC) spectrum well that were first calculated from the Wyoming OPC measurements when looking at bimodal PSD data. For the AEC that may be true, however there will very likely be a much stronger bias in the three SLN PSD parameters (median radius, sigma and number density) individually (see for example von Savigny and Hoffmann (2020), which you also cite). There would likely be an especially strong effect on the number density that is stronger the larger the second mode of the bimodal "true" distribution is. This in turn would introduce possibly large errors into other quantities that you calculate, like the surface area density (SAD). Please discuss the possible effects of a wrong SLN assumption in the main text, as it is central to the retrieval method.

This is an important point, thanks for mentioning this. We agree that it belongs in the main text and add a short discussion in the conclusion to highlight this, including the reference to von Savigny and Hoffmann.

**Reviewer 2:**

The stratospheric aerosol layer is a key element of the Earth's climate system through its impact on radiation, climate and heterogeneous ozone chemistry. Satellite observations provide global but limited information on the overall aerosol optical and microphysical properties needed to derive their impacts on radiation and chemistry. While modelling stratospheric aerosol has improved over the past decade, it is still important to derive prescribed stratospheric aerosol properties for models who do not simulate them directly. In this paper, Jorimann et al. (2025) describes a dataset which has been used in Climate Model Intercomparison Project (CMIP) and Chemistry Climate Model Initiative (CCMI) projects since 2011 under several names. While other forcing datasets might be available for future CMIP7, it is extremely important to fully understand the differences that may arise from utilizing different approaches. Overall, the paper is well written, clear and needs only a few modifications before it can be published in ACP.

We thank the reviewer for the insightful suggestions and comments and address each of them individually.

- Page 4, line 104.

Cloud-clearing data near the tropopause is a very important step before providing a cloud-free dataset. However, this manuscript provides only one sentence about this step which is also not correct. I do not understand why the author speaks about a "manual step" here. Several approaches have been taken to cloud-clear SAGE data and published in the literature for several decades but this is only recently that an official approach was included in the SAGE III dataset known as SAOTCM (Kovilakam et al., 2023). Other methods from Thomason and Vernier. (2013) and Bhatta et al. (2023) have been proposed. I do not think that a manual-based only approach has ever been conducted to cloud clear SAGE data. This sentence should be changed and corrected.
**Surendra Bhatta, Amit K. Pandit, Robert P. Loughman, and Jean-Paul Vernier, "Three-wavelength approach for aerosol-cloud discrimination in the SAGE III/ISS aerosol extinction dataset," Appl. Opt. 62, 3454-3466 (2023)**

We agree with what the reviewer wrote about cloud-clearing. In fact, it is exactly in line with the meaning of our text. Prior to these more recent methods that were proposed, various other ways - that we called "manual" - were employed. We understand that this expression could be misunderstood and thus changed the wording.

- Incorporation of OMPS into existing climatology. Page 4 line 110-114

While this is not the main point of this paper, the idea of incorporating limb scatter measurements into existing climatology is important (e.g. CREST). It would be important to improve the discussion on what might be the pros and cons of doing that. For example, OMPS extinction product requires a priori assumptions on stratospheric aerosol size distribution.

We discuss this aspect in "3.2 Wavelength quality", where we mention that a priori assumptions, as well as processing steps like transposing wavelengths, might limit the information content of data sets. Since every data set is different, we decided not to discuss this for any single data set like OMPS or CREST, but to structure the text into a guide (and record of our work at the same time) to check the quality of each channel. We believe this is the most useful information for any user and the best way to advise readers that these different challenges related to the data set type exist. To make the paragraph clearer we add one sentence to relate the issue of a priori assumptions specifically to limb scatter measurements.

- Page8-line195-"an program" should be changed to "a program"

Corrected, thank you.

- Page 9-figure9.

Figure 9 shows "strange" structure with a distinct rise error at 27 km. Why does the error come from GloSSAC since the title says that the mean error comes from SAGE III/ISS and its 6 wavelengths. Please clarify. Is there a potential explanation for this artefact at 27 km ?

We assume the reviewer is referring to Figure 2 here, which shows a latitude-altitude plot of the error. The reason for the rise is briefly mentioned in the text and figure caption: GloSSAC includes a high-altitude climatology, which starts at 27 km and replaces any transient measurements. We expand on the relevant sections to make this statement more clear. We recognize, however, that the description was lacking, since both a retrieval based on 6 SAGE III wavelengths and GloSSACv2.22 were mentioned. We add the specification that GloSSACv2.22 data were used, wherever 6 SAGE III/ISS wavelengths were not available. This is mostly the case at high altitudes, hence the rise in error associated with the high-altitude climatology.

- A new SAGE product (V6) is now available and ozone cross-section corrections should improve the mid-visible channels that have some issues and were not used (e.g. Figure 6). Could you comment on that?

We thank the reviewer for this point. Updates like this one might certainly improve the wavelength quality and new and updated data should be continually assimilated and used to retrieve the latest aerosol properties. We add a statement at the end of the conclusions to highlight this.

- I should say that my most significant comment would be in the validation section and especially the comparison between REMAP and POPS (Fig. 10). I spent time trying to understand this figure and I would first suggest some improvements. First since the figure uses 2 x axis (top and bottom), I would suggest coloring the axis according to the color of the line plots and use dashed and continuous line for REMAP and POPS. In addition, the legend could also contain the name of the variable displayed (extinction or SAD). If I read this plot correctly, it looks like REMAP does not agree well with POPS. First, the peak in SAOD and Extinction do not really match but are apart by several kilometers. Do you have an explanation on why it might be the case? In addition, the structure of the plume is rather different. Invoking the difference in sampling might be an explanation but to me it is not the only problem. I suspect that additional work might be needed to improve the comparison and provide further explanations about the differences observed.

*Concerning Figure 10 and the REMAP-POPS comparison*:

We deeply appreciate the reviewer's critical perspective on the comparison between REMAP and POPS (balloon-borne measurement). To understand the comparison better and expand our reasoning, we have re-examined this part and identified a mistake in the data analysis that caused a mismatch in latitude (not actually altitude, but to a similar effect). Correcting this mistake in the comparison now yields much better results. We revised Figure 10 and Figure S4 and changed the text describing the figure and text related to this comparison in the conclusion. We provide some extended discussion around the key points here:

**The role of temporal averaging**
POPS captures momentary states of the aerosol and its vertical SAD profiles thus come with some uncertainty (besides the inherent instrument uncertainties). Its data are subject to short-term variability. This can be understood and visualized using both the balloon ascent and descent data (previously we

used ascent only). We find that on some days the variability (or measurement uncertainty) is much larger than on others. The REMAP data consistently agrees much better with POPS, when the ascent and descent lines fall close together. Here, it is worth reiterating that REMAP operates on a data set that provides monthly mean values, where transient structures will be smoothed out. We assume that some of the bumps and peaks that POPS shows are lost in the averaging process, which only preserves robust structures.

**The role of zonal averaging**

GloSSAC (which was used for the retrieval) reports zonal means for latitude bands of 5° breadth. To examine the role this additional mode of averaging (over all longitudes), in **Figure A** below we provide single occultation profiles retrieved by SAGE III/ISS:

- 3 taken within 2 hours of the balloon measurement in a similar latitude (44-46 °S) but at different longitudes (174-80 °E)
- and 1 taken some 5 hours before the balloon measurement but only ~100 km from the balloon launch site

These are raw data of extinction coefficients at a single wavelength, rather than SAD. However, we think that the 4 profiles all show visibly different data below 20 km altitude, either on account of measurement uncertainty (high, integrated slant path optical depth and limited dynamic range of detector) or changing states of the lower stratosphere with changing meridional location.

Beyond those sources of differences, we agree with the reviewers that additionally there could be other reasons for disagreement. Notably, we acknowledge the notion that POPS might resolve a second mode (especially in volcanic conditions), whose SAD REMAP could underestimate, because the best solution for fitting AEC might not always yield the correct size distribution.

We adjust the text to elaborate on the comparison between REMAP and POPS according to our improved understanding. We also incorporate the reviewer's style suggestions to improve Figure 10.

[Figure]

**Figure A**: This figure shows the event physically closest to the balloon measurement (17:22) and 3 others at similar latitudes around the time the balloon measurement was performed in Lauder NZ (~22:00-24:00). There is variability of an order of magnitude in aerosol extinction below 20 km altitude (without outliers).